# Orthopoxvirus Zoonoses—Do We Still Remember and Are Ready to Fight?

**DOI:** 10.3390/pathogens12030363

**Published:** 2023-02-21

**Authors:** Małgorzata Gieryńska, Lidia Szulc-Dąbrowska, Justyna Struzik, Karolina Paulina Gregorczyk-Zboroch, Matylda Barbara Mielcarska, Felix Ngosa Toka, Ada Schollenberger, Zuzanna Biernacka

**Affiliations:** 1Department of Preclinical Sciences, Institute of Veterinary Medicine, Warsaw University of Life Sciences—SGGW, 02-786 Warsaw, Poland; 2Department of Biomedical Sciences, Ross University School of Veterinary Medicine, Basseterre P.O. Box 334, Saint Kitts and Nevis; 3Doctoral School, Warsaw University of Life Sciences, 02-786 Warsaw, Poland

**Keywords:** orthopoxvirus zoonoses, monkeypox, immune response, immunological memory, immunoprophylaxis

## Abstract

The eradication of smallpox was an enormous achievement due to the global vaccination program launched by World Health Organization. The cessation of the vaccination program led to steadily declining herd immunity against smallpox, causing a health emergency of global concern. The smallpox vaccines induced strong, humoral, and cell-mediated immune responses, protecting for decades after immunization, not only against smallpox but also against other zoonotic orthopoxviruses that now represent a significant threat to public health. Here we review the major aspects regarding orthopoxviruses’ zoonotic infections, factors responsible for viral transmissions, as well as the emerging problem of the increased number of monkeypox cases recently reported. The development of prophylactic measures against poxvirus infections, especially the current threat caused by the monkeypox virus, requires a profound understanding of poxvirus immunobiology. The utilization of animal and cell line models has provided good insight into host antiviral defenses as well as orthopoxvirus evasion mechanisms. To survive within a host, orthopoxviruses encode a large number of proteins that subvert inflammatory and immune pathways. The circumvention of viral evasion strategies and the enhancement of major host defenses are key in designing novel, safer vaccines, and should become the targets of antiviral therapies in treating poxvirus infections.

## 1. Introduction

Understanding the interactions between immune cells is the first step to revealing the immunological mechanisms involved in host protection when facing an infectious disease. This knowledge makes it possible to design effective and safe vaccines, the ultimate tool for controlling viral diseases in humans and animals. More than 220 years ago, Edward Jenner announced his achievement of prophylactic immunization against smallpox, the most devastating disease of mankind [1].

The original Jenner vaccine contained the cowpox virus (CPXV) strain, which cross-protected man against smallpox without causing the disease in vaccinated individuals. Designing a protective vaccine was the first step that made it possible for World Health Organization (WHO) to declare in 1980 that smallpox was eradicated.

Smallpox, a serious, highly contagious disease characterized by high morbidity and mortality, and one of the most devastating diseases that had plagued humanity for centuries, was caused by *Variola virus* (VARV), a member of the virus family *Poxviridae*, subfamily *Chordopoxvirinae*, genus *Orthopoxvirus*. Irrespective of VARV, the genus *Orthopoxvirus* includes *Akhmeta virus*, *Alaskapox virus* (AK2015-poxvirus), *Buffalopox virus* (BPXV), *Camelpox virus* (CMLV), *Cowpox virus* (CPXV), *Horsepox virus* (HSPV), *Monkeypox virus* (MPXV), and *Vaccinia virus* (VACV), which can all cause variola-like zoonoses in humans, as well as several more animal poxviruses (*Ectromelia virus* [ECTV], *Rabbitpox virus* [RPV], *Taterapox virus* [GBLV], *Volepox virus* [VPXV]), and several others [2,3]. Viruses belonging to *Poxviridae* are large, brick-shaped, or ovoid virions with a double-stranded DNA (dsDNA) genome. Their replication occurs in the cytoplasm of the infected host cell. Their genome consists of a single, linear molecule of dsDNA that is covalently closed at each end, amounting to approximately 128 to 365 kbp (genera *Parapoxvirus* and *Avipoxvirus*, respectively), which codes for over 200 genes, including genes that encode proteins involved in virus evasion in innate and adaptive host immune responses. These poxvirus proteins can affect the recognition of viral infection, influence intracellular signaling pathways, inhibit the activation of transcription factors as well as adaptor proteins, and, by the latter, develop evasion strategies.

During infection, orthopoxviruses (OPXVs) can be detected in two, antigenically distinct forms: designated mature virions (MV) or extracellular enveloped virions (EV). The MV form contains more than 80 proteins and consists of a nucleoprotein core surrounded by a lipid membrane with approximately 20 proteins. These core proteins are responsible for the synthesis and modification of viral mRNA. The EV form consists of the MV form covered by the supplementary membrane presenting about 9 proteins distinct from those in the MV membrane: A33, A34, A36, A56, B5, E2, F12, F13, and K2 [4,5,6,7,8,9,10,11,12,13]. Serologically, all OPXVs share some epitopes, therefore, infection with any representative of the genus *Orthopoxvirus* results in the induction of cross-protective immunity against other members of this genus [14,15].

The smallpox clinical picture is characterized by a maculopapular rash that converts into vesicles, then ulcers, which become scabs, leaving characteristic, deep scars—pocks. VARV is resistant to drying, so it persists in the infectious status in scabs and epidermal cells, as well as in patients’ bedding and clothing for an extended time. The virus can be transmitted *via* the respiratory route with sputum and saliva droplets that have a range of likely no more than 2 m. In such a case, there is a threat to those in the immediate vicinity of the affected patient and also those in close contact with the skin, body fluids, and ulcer/scab content, which are the source of the infectious agent. The incubation period of smallpox is approximately 7 to 19 days, after which the systemic spread of the infection by the macrophages and lymphocytes occurs. A high fever accompanies the primary viremia, and, usually after 12–14 days, the characteristic skin lesions appear. The general clinical signs include malaise, chills, severe headaches, muscle pain, weakness, abdominal pain, and vomiting. From the beginning of the clinical period through the recovery phase, the patient continuously releases VARV, and the smallpox victim remains the source of infection for at least 3–4 weeks [16,17].

In 1966, WHO, to combat the disease, introduced the smallpox eradication program—SEP (1966–1980). The last naturally occurring case of smallpox was noted in 1977 in Somalia. Thus, in 1980, following the global campaign of surveillance and vaccination, WHO declared smallpox eradicated, the only human infectious disease that has achieved this distinction. Smallpox eradication was facilitated by the fact that VARV affected only humans and had no natural vector, reservoir, or carrier, and furthermore, that recovery leaves the patient immune for life. However, the eradication of smallpox caused the cessation of the global surveillance and vaccination program. More than 40 years after this act, we face the consequences in terms of the increasing population without immunity to smallpox and also to some other zoonotic OPXVs, since vaccination against smallpox had also provided protection, at least partially, against them [2,14,18,19,20,21,22,23,24].

Importantly, it must be emphasized that the infectious agent responsible for smallpox still exists. There are two WHO-designated sites where stocks of variola virus are stored and used for research: Centers for Disease Control and Prevention, Atlanta, Georgia, United States, and the Russian State Centre for Research on Virology and Biotechnology, Koltsovo, Novosibirsk Region, Russian Federation [25]. Likewise, over the years, there has been a rapid development of molecular biology techniques, and now, certain achievements that have been impossible in the last century are within reach. For instance, in 2018, infectious HSPV was created from chemically synthesized DNA fragments to serve as a novel replication-proficient smallpox vaccine. Therefore, there is potential for the development of a new form of VARV or another virulent OPXV that is pathogenic for humans. Due to the large size of the poxvirus genome, it is quite possible, through homologous recombination, to insert genes encoding drug-resistance, which will lead to the increased virulence of the virus, or to insert genes encoding immunoregulatory mediators that will affect the immune response in the infected host, as it was done with ECTV by inserting the gene encoding IL-4 [16,26,27]. The aim of this manuscript is summarized in Figure 1.

## 2. Orthopoxvirus Zoonotic Infections

Zoonosis is an infectious disease that is naturally transmitted between species, from vertebrate animals to humans, and may present a significant risk to public health. Within the genus *Orthopoxvirus,* zoonotic species can be distinguished, among which the most important are BPXV, CMLV, CPXV, HSPV, MPXV, and VACV. Traditionally, the name OPXV species was given based on the animal species from which the virus was isolated. However, the name does not necessarily reflect its natural reservoir or the complete range of the hosts. The origin and evolution of poxviruses are still not fully elucidated. The data regarding the complete nucleotide sequences in various strains showed that the evolution of the poxvirus genomes occurred through the gaining and losing of genes, usually through duplication/amplification and horizontal gene transfer. As a result, there is a presence of some genes that are intact in one species but fragmented or deleted in another [14,15,28]. There are genes called host range genes, and they are responsible for the pox-virus-specific differences in tropism and host range, allowing viral replication only in a set of cell lineages that originate from specific tissues or host species. Also, many of the genes are not required for viral replication under *in vitro* conditions; nevertheless, they are essential to the modulation of host antiviral protective immune response, therefore, they are considered as virulence genes [14,29,30]. Among the genus *Orthopoxvirus*, CPXV was proposed as a progenitor virus for all members of this genus since it has the largest genome, containing all the genes found in other OPXV species, in contrast to VARV, which has the smallest genome of all OPXVs as the result of multistage reductive evolution [15,28].

Following the cessation of the global vaccination program, a steadily increasing number of OPXV infections responsible for zoonoses in humans is being observed, and is it strongly connected with waning smallpox immunity. The circulation of poxviruses among wild and domestic animals has been recorded in various areas of the world, among which are Europe, the Middle East, Asia, Africa, and South America [2,20,21,31,32]. This raises concerns about the possibility of a sudden outbreak that may result in a severe impact on public health. Additionally, other factors are involved in increasing the prevalence of poxvirus incidents in humans. Despite their historic notoriety, infections caused by poxviruses are commonly missed in today’s world and are most often not properly diagnosed. In humans, zoonoses caused by poxviruses are often neglected, and undoubtedly, the data on these infections are incomplete, therefore, the actual epidemiological picture is unknown. The import of exotic animals, considered companion animals and livestock, has contributed to the spread of poxviruses, such as CPXV, MPXV, novel VACV as well as BPXV and CMLV, due to coming into close contact with nonimmune native animals, and humans. Furthermore, the transportation and abandonment of infected companion animals were responsible for the introduction of zoonotic poxviruses to areas outside their normal geographic range [15,33,34]. That is why the awareness of owners and veterinarians about the possibility of poxvirus infections in companion animals is so important. In the past, recognizing skin lesions caused by smallpox was relatively easy for the experienced clinician. Nowadays, healthcare professionals and veterinarians must be promptly educated about the characteristics of OPXV infections. A delay in the correct diagnosis of the animal infection may lead to the inadvertent exposure of the owner. Additionally, the correct diagnosis is necessary to introduce appropriate procedures to prevent the further spread of the virus. Infections with OPXVs in domestic and companion animals may be asymptomatic, sub-clinical, or may take the form of a skin rash with progressive respiratory signs leading to death [35,36]. The schematic transmission of OPXVs to humans is presented in Figure 2.

### 2.1. Cowpox

Cowpox is caused by CPXV, the virus that was used for the first time by Edward Jenner to induce protective immunity against smallpox. Bovine cowpox, the transmission of which occurs from cow to man, was reported in Europe, mainly in those remaining in close contact with infected cows. Nowadays, the virus can infect a very broad range of animals including non-human primates, felines, dogs, rodents, foxes, rhinoceroses, tapirs, okapis, horses, anteaters, mongooses, stone martens, bearcats, and farmed llamas, although most of these animals are thought to be accidental CPXV hosts [37]. Currently, infections with CPXV are reported mostly in Europe and Western Asia, and rodents, particularly, voles (*Microtus* spp. and *Myodes* spp.) are considered the major reservoir of CPXV in nature. However, depending on the region, wood mice (*Apodemus sylvaticus*), bank voles (*Myodes glareolus*) (the UK), as well as laboratory mice and wild rodents (Turkmenistan and Russia) can serve as a source of CPXV [20,21,32,38]. Human CPXV infections have been connected with cattle. Additionally, domestic cats are pointed out as another source of CPXV infection since they get infected due to their hunting behavior and often come into contact with infected rodents [31,39]. Feline infections are correlated with the peak population size and activity of rodents, therefore, infections occur during hunting, through scratches, bites, and, probably, ingestion. The first domestic case of cat-to-human CPXV transmission was reported in 1985 [40]. The transmission of the virus to humans takes place through direct contact with the infected animal, as well as its body fluids. Scratches and abrassions of the skin indicate the localization of further clinical changes, and it seems that children and people involved in animal care appear at the highest risk [41,42,43,44].

The presence of pet rats (such as *Rattus norvegicus*) and other non-domestic animals, especially exotic species treated as pet animals, raises concerns regarding the emergence of cowpox, including in new geographical regions [2,32,36,43,45]. In humans, clinical cowpox lesions located in the skin, and sometimes mucosa, are large and ulcerative with inflammation and edema. The formed crust is thick, hard, and black. Additionally, local lymphadenopathy as well as pyrexia, lethargy, sore throat, and general malaise can be observed. These lesions can be secondarily infected and form scars. The disease is self-limiting in immunocompetent patients, although it can be generalized and severe in patients with immunodeficiencies. Clinical cowpox in humans can be mistakenly diagnosed as herpes, parapox, or anthrax [20,35,44].

### 2.2. Vaccinia Virus

VACV is the most studied and the most mysterious virus belonging to the genus *Orthopoxvirus*. The attenuated strains of this virus were used worldwide during the smallpox eradication program launched by WHO, but the origin of VACV was for many years unknown. Recently, due to the sequencing of the complete genome of HSPV, it was revealed that there is a close relatedness between those two viral species. There is a strong suggestion that VACV could have been derived from zoonotic HSPV since Jenner indicated the genesis of his vaccine from the skin infection of the heels of horses, known as grease heels, and to make the vaccine against smallpox more suitable for humans, he passaged it through the cow [15]. Even though the role of VACV in combating smallpox was proven, severe, post-vaccinal complications have been noticed, including progressive vaccinia, generalized vaccinia, vaccinia eczema, sometimes encephalitis, myocarditis, or ocular complications. The frequency of adverse effects has depended on the vaccine strain [46].

For many years it was assumed that, due to laboratory attenuation, VACV is not capable of establishing a natural reservoir in the environment. Nevertheless, various strains of VACV were isolated from different parts of the world, indicating their ability to adapt to new hosts and environments [21,47,48,49,50].

Currently, various s trains of VACV circulate in South America, mainly in Brazil, Uruguay, Colombia, and Argentina, causing outbreaks in dairy cattle and affecting dairy handlers. Since 1999, an increasing number of exanthematous disease cases caused by VACV was noticed. The disease bovine vaccinia in cows causes a form of lesions to develop on their udders and teats characterized by papules that evolve into ulcers. Dairy workers handling such infected animals develop pleomorphic lesions on their hands (mainly papules and painful ulcers), fever, and lymphadenopathy, however, nausea, vomiting, diarrhea, mental confusion, and seizures have been reported too [15,21,31,47,51]. The history of VACV introduction to Brazil is complicated since the first human smallpox vaccine was imported by the arms of slaves returning from Portugal (1804), then at the end of the 19^th^ century, the first animal vaccine produced in calves was imported and then distributed in a different area of the country, introducing the government vaccination program against VARV [47,52]. It developed into well established Brazilian National Immunization Program providing protective vaccination not only against VARV but also against other preventable diseases like poliomyelitis, rubella, and rubella-syndrome, as well as diphtheria, tetanus, and neonatal tetanus. Nowadays, however, in the context of OPXV infections, new problems have emerged, and they are strongly connected with the misinformation regarding OPXV infections and the stigma associated with MPXV infection, as well as health care system funding [52,53,54].

Thus far, several VACV strains have been identified and named according to their geographical origin; moreover, their influence on farm animals other than cattle, as well as on wild animals, has been well documented. Nowadays, bovine vaccinia (BV) is considered an endemic disease. The issue regarding the origin of the infectious agent responsible for bovine disease remains unsolved. One hypothesis considers the pathway from vaccinated humans to domestic animals, and further to wild animals, with subsequent adaptation. Another hypothesis proposes the natural genetic and phenotypic diversity of VACV strains that circulate in a so-far unknown reservoir. An additional hypothesis that cannot be excluded, proposed by Sergei Shchelkunov, focuses on the origin of VACV. Since several strains of VACV have been isolated from cows, horses, humans as well as rodents, and since there is close relatedness between HSPV and VACV, the viral strains could have been accidentally exported several times from Europe to South America through infected horses and/or rodents and then introduced into the native wildlife. A recent phylogenetic analysis of VACV isolates from various states of Brazil suggested that the origin of BR-VACV was independent of vaccine strains used during the 20th century [15,47,49,55,56,57]. Nevertheless, VACV strains established their presence in the environment with a very wide host range, therefore, even though the data regarding the frequent transmission of the virus between wild and domestic herbivores are scarce, the growing concern for humans is real. Serological studies, the usual method for monitoring the prevalence of infectious disease, performed in rural populations of North Brazil, revealed a 27.9% seroprevalence against OPXVs in patients. Among those OPXV-seropositive persons, 23.4% were not vaccinated against smallpox, suggesting that they were naturally exposed to circulating OPXVs [58]. Cattle likely play a major role in the transmission chain of VACV. The presence of VACV in milk and feces favors the maintenance of the virus in the environment and has been well documented. Moreover, dairy products should be considered an alternative route for VACV spreading, since infectious particles of VACV have been isolated from cheese prepared from milk experimentally contaminated with the virus, after 60 days of ripening [21,59,60]. Later, the participation of other farm animals in the dissemination of VACV was proposed, although the direct transmission of the virus to humans has yet to be documented. Finally, the interactions between wild rodents that could be VACV reservoirs and peridomestic rodents serving as the bridge for spreading VACV strains between wild and rural environments can promote transmission among farm and wild animals. Studies done by Siqueira Ferreira et al. focused on explaining the ability of OPXVs to survive in the environment. Since OPXVs can replicate in the intestinal epithelial cells, feces can be a source of VACV, therefore, the presented murine model can play a significant role in understanding VACV spread and transmission [21,61,62].

Bovine vaccinia is a self-limiting disease in immunocompetent individuals but it can affect immunocompromised patients, presenting a severe, generalized disease, similar to the cases of human cowpox. It should be emphasized that once cattle, as a reservoir of CPXV, were replaced by cats, an increased number of human cowpox cases was observed. A similar scenario should be considered in the case of VACV since the two viruses present many similarities and are characterized by a wide host range [20,21].

### 2.3. Zoonotic Outbreaks Caused by CMLV and BPXV

A phylogenetic analysis of OPXVs revealed strong closeness between BPXV and VACV as well as CMLV and VARV. Indeed, CMLV is genetically closest to VARV, the agent of smallpox, amongst any of the other OPXV species, and carries genes related to host immune evasion. It may hypothetically facilitate the reintroduction of VARV, but it is of highest concern to public health. On the other hand, BPXV is a close variant of VACV, the Lister vaccine strain, which had been originally used for inoculating buffalo calves to produce a smallpox vaccine [15,21,63,64,65]. The phylogenetic tree of OPXVs is updated on regular basis and provided by the International Committee on Taxonomy of Viruses [3].

The discovery of CMLV and BPXV was achieved around the time of the smallpox epidemics and the beginning of the vaccination programs with VACV. Camelpox is one of the most important viral, host-specific diseases in camels. It is highly contagious and has a serious impact on the health and mortality of camels, causing pox-like lesions localized on the head, neck, inguinal regions, and other hairless areas of the body. Similar lesions occur on the mucus membrane. The disease is reported in camel-rearing belts, particularly in Africa (north of the equator), the Middle East, and Asia. CMLV is host-specific and does not infect other animal species like cattle, sheep, and goats. The transmission of the disease occurs either by direct contact between infected and susceptible animals or by indirect infection *via* contaminated fomites. Bera et al. reported the first zoonotic CMLV infections in humans in 2009, which were associated with camelpox outbreaks in dromedary camels in the northwest region of India. The zoonotic cases were diagnosed in animal handlers and attendants, with skin lesions characteristic of poxvirus infections such as papules, vesicles, ulceration, and scabs on the fingers and hands [64].

BPXV was found to be derived from VACV, the Lister vaccine strain, and this supposition was supported by an analysis of BPXV’s complete genome sequence [66,67,68]. The appearance of BPXV in the environment occurred through the gradual adaptation of the vaccine strain until it converted to an infectious, fully virulent virus. BPXV was first described in 1934 and subsequently isolated in 1967, and in the same year, buffalopox was recognized as an important zoonotic disease [65,68]. The disease caused by BPXV affects domestic buffalo, cattle, and humans who are in close contact with infected animals, and resembles the major characteristics of VACV. It is a highly transmissible zoonosis that has persisted for over 40 years following the cessation of global vaccination against smallpox. The disease is manifested by a high fever and severe local skin edema, which are often associated with the eruption of pox lesions. The skin lesions in animals are limited to the udder and teats, leading to mastitis, although in dairy workers who get infected accidentally, pox-like lesions occur on their hands, forearms, and foreheads. is The lesions are accompanied by fever, axillary lymphadenopathy, and malaise [2,31,65,66].

In the years 2004–2005, a nosocomial outbreak of BPXV occurred in Karachi, Pakistan, as a result of using fat from BPXV-infected buffalo as “first-aid medication” in the treatment of skin burns [69]. Although a variety of animal species, such as guinea pigs, BALB/c, Swiss mice, cows, buffalo calves, rabbits, and chickens, seem to be susceptible to experimental BPXV infection, the role of those species in BPXV transmission, dissemination, and maintenance in the environment still requires elucidation. Up to now, human-to-human transmission of BPXV has not been documented [21].

Although zoonotic cases caused by CMLV and BPXV are sporadic, it is important to understand the immunobiology of these viruses since the majority of the world’s human population has not been vaccinated against smallpox. CMLV is most closely related to VARV and, similarly, it encodes many genes, such as chemokine-binding proteins, TNF Receptor II crmB, complement binding proteins, protein kinase inhibitors, signal transducers and activators of transcription (STAT) 1-inhibitor, serine proteinase inhibitors, CD47-like proteins, IL-1/Toll-like receptor inhibitors, IFN inhibitors, IFN-γ receptors, and IFN-α/β binding proteins, which allow immunomodulation and the escape from host protective defense mechanisms [63,70,71,72,73]. It has to be indicated that those evasion pathways are not unique only to CMLV but also to other OPXVs. In the case of BPXV, we are dealing with a virus that has evolved in a relatively short time into an infectious agent, derived from a strain believed to be unable to establish an animal reservoir in nature. The possibility of BPXV and CMLV infections raises concern for global health since introducing these viruses to new geographical areas is quite possible due to many factors, among which the trade of animals or animal-derived products may play a key role.

### 2.4. Monkeypox

The recent emergence of numerous incidents of monkeypox as well as the possibility of the human-to-human transfer of MPXV posed a question regarding the safety of humans in the context of waning immunity to OPXV infections as a result of SEP cessation. Infections caused by MPXV were first reported in 1958 by Magnus et al. in Copenhagen, Denmark, when the pustular rash illness was observed in cynomolgus macaques (*Macaca fascicularis*) transported from Singapore [21,74,75,76]. In the following years, more outbreaks in USA and Netherlands were identified in captive monkeys transported from India, Malaysia, and the Philippines [77]. Although the name of the causative agent responsible for the clinical skin lesions comes from the species that was first diagnosed with the infection, non-human primates are accidental hosts for MPXV; furthermore, humans are also incidental hosts for infections caused by this virus.

Before the eradication of smallpox, human MPXV infections were probably misdiagnosed as VARV infections because of the high prevalence of smallpox and similarities in the progression and clinical manifestations of both diseases. Human monkeypox was first recognized as a disease that is different from smallpox in 1970 in the Democratic Republic of Congo (DRC) in a 9-month-old child who presented smallpox-like skin changes. This was 2 years after the last case of smallpox in this area. The development of a rash, with subsequent crusting, was observed. The child, who had never been vaccinated against smallpox, recovered. However, despite the initial recovery, the patient succumbed to measles and died [78].

Monkeypox is a zoonotic disease that occurs endemically in Central and West Africa but is mostly concentrated in the DRC, and it has previously been rare outside of this area. The clinical manifestations are similar to smallpox but with fewer fatalities, although the disease severity depends on the virus strain. MPXV shares a high level of sequence similarity with VARV (96%); however, as indicated earlier, VARV is most related to CMLV. Historically, MPXV was divided into two distinct clades, namely the Congo Basin (Central Africa) clade and the West African clade. However, with the current situation, a new classification of MPXV has been proposed: clade 1 and clade 2 represent the Congo Basin and the West African clades, respectively, and clade 3 represents outbreaks outside of Africa. The genetic differences between clades have provided clues to understanding the differences within human monkeypox disease pathology [79,80,81,82,83]. Indeed, the Congo Basin clade, found mainly in the DRC and neighboring countries, is more pathogenic, with a higher fatality rate (11%) than the West African clade (1%), and has been documented in cases of human-to-human transmission [74,84,85,86].

Studies focused on finding the major endemic origin of MPXV infections have pointed towards certain sylvatic rodents as the largest reservoirs of MPXV, although it is still not clear which of them serves as the utmost source. Despite many efforts, up to this point, MPXV was reported to have been isolated only twice from wild animals that were either ill or dead presenting pox-like lesions. The first isolation was from a rope squirrel (*Funisciurus* spp.) in the DRC in 1985, and the second one was from a sooty mangabey (*Cercocebus atys*) in Taï National Park, Ivory Coast, in 2012. Serological surveys have revealed a high seroprevalence of anti-OPXV among animals. Needless to say, seropositivity may be the result of infections of unidentified OPXV species. However, ecological surveys have reported that rope squirrels, sun squirrels (*Heliosciurus* spp.), pouched rats (*Cricetomys* spp.), and dormice (*Graphiurus* spp.) could be natural reservoir hosts since they were found to have the highest frequency of OPXV seropositivity. MPXV outbreaks were reported among populations of chimpanzees in Taï National Park, in 2017, and among captive chimpanzees housed at wildlife sanctuaries in Cameroon in 2014 and 2016. Further research performed on laboratory or captive animals has indicated a very wide range of animal species susceptible to MPXV infection. Among others, a broad range of rodents (e.g., mice, rats, rabbits, hamsters, jerboas, and black-tailed prairie dogs), ant-eaters, southern opossums, African hedgehogs, non-human primates of the Old World as well as New World, and many other species are susceptible to the challenges of MPXV [21,87,88,89,90,91,92,93].

MPXV, endemically, is transmitted through direct contact (such as a scratch or bite) with infected live or dead animals, together with their bodily fluids, respiratory droplets, and lesion exudates, as well as crust materials. It is worth noticing that rodents like squirrels and giant pouch rats, considered the main reservoir of MPXV, are the source of the nutrients. As a consequence, apart from contact with live animals, handling rodents and preparing or consuming wild game or bushmeat poses the risk of contracting MPXV [84,94,95].

Human-to-human transmission is relatively inefficient, although incidences of the disease have dramatically increased in recent years [2,74,84,96,97]. It can occur by respiratory droplets or by direct contact with infected lesions or bodily fluids as well as crust material that can infect the mucus membranes of the respiratory tract, conjunctiva, or wounded skin. Furthermore, mother-to-child MPXV transmission can occur through the placenta during pregnancy, leading to the death of the fetus (congenital monkeypox) or the newborn baby, through skin contact during or after birth. Though prolonged close physical contact is required for monkeypox transmission, it is not clear if the virus can be transferred through sexual routes [95,98].

As it was indicated, MPXV can enter a host through the oropharynx, nasopharynx, or intradermal routes. Human monkeypox, which is similar but milder to smallpox, can be described with three distinct phases: incubation, prodrome, and rash. Following entry, the virus replicates at the portal of entry, then disseminates to the regional lymph nodes, followed by systemic viral spread. This represents the incubation period and generally lasts 6–14 days with an upper limit of 21 days [84,90,99,100,101]. The prodrome phase usually lasts 1–2 days and is characterized by “flu-like” symptoms, including fever, headaches, fatigue, muscle soreness, and enlarged lymph nodes. Lymphadenopathy (particularly in the cervical and inguinal regions) is a feature that distinguishes monkeypox from smallpox and chickenpox (caused by a varicella-zoster virus), two diseases with which monkeypox was initially confused. Subsequently, the rash develops with a distinct pattern. It starts as a macular rash followed by a papular, vesicular, and pustular rash before crusting over and falling off. Initially, the rash appears on the face, then spreads to the trunk and extremities (including the palms and soles) and is centrifugally concentrated. It can also be visible in the mouth, conjunctiva, and cornea, as well as on the genitalia [85,90,94,99,100,102,103,104].

During the rash phase, lesions that are in the same stage of development contain infectious viral particles, which can be transmitted to the susceptible recipient through direct contact.

Monkeypox is a disease with mild symptoms that self-resolves within 2–4 weeks with complete recovery. However, secondary complications may occur as a result of bacterial infections. These comprise bacterial skin infections, gastroenteritis, sepsis, bronchopneumonia, and encephalitis. Another long-term complication, although it is rare, is the loss of vision resulting from cornea infection and the scarring of tissue [85,90,105]. Moreover, the severity of the disease’s symptoms depends on several factors, including the viral strain, route of infection, degree of virus exposure, the age, and immunological status of the patient. Severe cases of the disease with increased clinical changes (prolonged monkeypox, larger lesions, higher rates of secondary infections, and death) may require hospitalization, especially when MPXV infections occur in children or in patients with immune deficiencies. The case fatality rate regarding the Congo Basin clade is higher (up to 11%) compared to the West African clade (around 3.6%), although these values only apply to African outbreaks. Epidemiological studies have shown a higher susceptibility to MPXV infections among younger children, pregnant women, and those who were not vaccinated against smallpox [85,97,106,107].

Initially, MPXV infections among humans were restricted to the African continent; however, the number of them has been increasing since 1970, even though the morbidity rate could be underreported. The first report regarding zoonotic MPXV infection outside of the endemic area was documented in 2003 with 71 human cases identified in the Midwest states of the US. This outbreak was connected with MPXV-infected rodents imported from Ghana. Infected dormice, rope squirrels, and Gambian giant rats were housed together with native prairie dogs that were afterward distributed as pets. During this first outbreak, no human deaths were reported and phylogenetic analysis of the viral strain responsible for infections indicated the involvement of the West African clade [79,97,108,109,110]. Due to the fast diagnosis and isolation of the disease, this first, multi-state zoonotic outbreak in the USA, was short-lived, and transmission through the country ceased.

Unfortunately, due to many factors, such as the increasing intrusion of humans into wild habitats, the importation of animals for laboratory studies and as pets, as well as the increasing ratio of travel from endemic areas to MPXV-free regions, together with the constantly waning herd immunity to smallpox, there has been a growing number of human monkeypox incidences. The first observed incidents of human monkeypox after 2003 were reported in the UK in 2018 and 2019 in patients traveling from Nigeria, where the re-emergence of MPXV infections was revealed [111,112]. Currently, in 2022, there is the largest and most dispersed non-endemic monkeypox outbreak. Moreover, since high numbers of monkeypox cases have been observed, there is a strong indication that, for MPXV transmission, close, physical, and personal contact must occur, and not zoonotic infections. The process of spreading the disease was summarized in recently published papers [21,74,84,86,113,114].

The major events connected with OPXVs are summarized in Figure 3.

## 3. Monkeypox Prophylaxis

The monkeypox multi-country outbreak is a public health emergency of international concern (PHEIC). WHO issued a declaration regarding monkeypox prophylaxis, focusing on approved vaccines against monkeypox. However, the recommendations for vaccination are issued only for people who are at risk (cases of infection and contact with infected persons). Until now (December 2022), mass vaccination is not recommended.

The eradication of smallpox was possible due to the global vaccination program with the vaccine based on the VACV strains inducing cross-protection in humans. Historically, first-generation smallpox vaccines, including Dryvax, Aventis-Pasteur smallpox vaccine, Lister/Elstree, EM-63, and Temple of Heaven, were live, replicating VACV strains. All of them were highly effective, although some caused rare but potentially life-threatening adverse effects in immunocompromised people, persons with certain skin conditions, cancer patients, organ transplant recipients, and people with heart disease. This eventually led to the cessation of using those vaccines among the general public and to the development of novel, prophylactic, or therapeutic vaccines. Designing a novel type of vaccine is vital since monkeypox has spread to multiple continents, from Africa to Europe and the Americas. As such, the vaccine should be available for prophylactic and therapeutic (post-exposure) use [115,116].

Furthermore, the first-generation vaccines also lacked standardization and control of the production process. Second-generation smallpox vaccines that were derived from the VACV strains used for the first-generation vaccines were highly improved with respect to these issues. The viral strains were propagated without genetic manipulations on cell culture under controlled conditions and were purified using the plaque-purification method. Vaccines such as ACAM2000 (derived from Dryvax) were prepared on Vero cells, and for Elstree-BN (derived from Lister/Elstree), chick embryo fibroblast cells were used [115,116,117]. Since the second-generation vaccine viral strains show close relatedness to the strains used for first-generation vaccines, they can induce a similar reactogenicity to the live, animal-derived vaccines. However, these replicating vaccines still showed unsatisfactory safety profiles [118]. Studies on the development of the third-generation smallpox vaccines began in the 1970s as the answer to the known adverse effects resulting from viral replication (generalized vaccinia, eczema vaccinatum, and post-vaccinial encephalitis). Parental strains were extensively passaged in cell cultures of various host origins. During this process, an alteration of viral characteristics occurred due to random genetic mutations. One of these improved vaccines is LC16m8—an attenuated, replicating, temperature-sensitive VACV strain derived from the original Japanese Lister strain [119,120]. Now, many research groups focus their work on the assessment of the immunobiology of OPXVs, and the data obtained by them can be used for the development of safer, immunogenic, and effective smallpox vaccines which design is based on the variety of proteins and/or peptide subunits and theOPXV DNA.

Currently, there are three available vaccines for OPXV infections, namely: ACAM2000, JYNNEOS (also known as IMVAMUNE, MVA-BN, IMVANEX), and LC16KMB (LC16m8), from which WHO recommends ACAM2000 and JYNNEOS for post-exposure monkeypox prophylaxis [16,120,121,122].

Since the eradication of smallpox, WHO has set tight regulations concerning studies using VARV, therefore, proper animal models should be used for designing and evaluating the potency of the new types of vaccines. The ideal animal model used for assessing the disease’s development, stimulating the protective immune response, and evaluating the efficacy of the new vaccine, should closely mimic the events induced during VARV infection. Yet, VARV is host-restricted to humans, and it was proven that very high virus doses are required to cause pathology in cynomolgus macaques used as the only alternative model [123]. Therefore, for many years, other animal models were engaged to recognize the immunobiology of OPXV infections, which is crucial to the development of safer vaccines and also targeted anti-viral therapies. The murine model was used to evaluate the immunobiology of VACV, ECTV, CPXV, and MPXV. Independent of the murine model, the impact of MPXV on the host was assessed using monkeys, dormice, ground squirrels, and black-tailed prairie dogs as animal models [115,124,125,126,127,128,129,130,131,132].

Since the incubation period of both VARV and MPXV is rather long, from 7 to 19 days and 6 to 14 days, respectively, host protection can be induced even when the smallpox vaccine is used after exposure to OPXVs. The information from the eradication time indicates that the prevention of mortality and the elimination or modification of morbidity caused by VARV infection can be achieved with the post-exposure vaccination introduced 1–3 days after exposure.

Second- and third-generation smallpox vaccines were not available when the SEP was introduced and conducted, therefore, their efficacy pre- and post-exposure to VARV was not directly evaluated. An assessment of the efficacy of ACAM2000 and JYNNEOS during the post-exposure period was conducted by Keckler et al. Black-tailed prairie dogs (*Cynomys ludovicianus*) were used as the animal model for studying respiratory challenges with MPXV. These animals, upon facing the systemic challenges of MPXV, presented the disease with an incubation period of 6–9 days, and then the clinical signs developed as localized primary lesions and disseminated secondary lesions. The development of the disease in the black-tailed prairie dogs resembled VARV infection in humans, and that made it possible to evaluate the pre- and post-exposure efficacy of the tested vaccines as well as the effectiveness of antiviral therapies. Furthermore, the similarity of the disease incubation period in this model to that of the development of human OPXV infections made it possible to extrapolate the results obtained from the post-exposure challenge [133,134,135,136]. In this study, ACAM2000 or JYNNEOS were administered to the black-tailed prairie dogs on day 1 or day 3 of post-exposure to the intranasal challenge with a low or high dose of the MPXV Congo Basin strain. The obtained results show the relevance of second- and third-generation vaccines as prophylaxes in MPXV post-exposure. Although the vaccines did not protect the prairie dogs when they were challenged with the high dose of the virus, the death of animals was significantly delayed compared to controls. In addition, the production of neutralizing antibodies was induced. While the level of protection was comparable to the controls in the case of the ACAM2000 vaccine on days 1 and 3 of administration, the protection was greater when the JYNNEOS vaccine was administered on day 1 than on day 3 post-exposure, even though JYNNEOS-induced antibody levels were as high as or even higher than ACAM2000. These data are significant for understanding the relevance of post-exposure vaccination. Especially, JYNNEOS, a third-generation vaccine, can be applied in persons who are contraindicated for ACAM2000, the replication vaccine [137].

The same animal model was used to evaluate the efficacy of humoral immunity induced with the above-mentioned smallpox vaccines. The animals were immunized twice, 60 and 30 (booster) days before the intranasal challenge, with a high or low dose of the MPXV Congo Basin strain. Vaccination with ACAM2000 and JYNNEOS resulted in the induction of a humoral immune response that was significantly boosted upon the MPXV challenge. Moreover, induced humoral immunity closely resembles the immunity in humans induced by vaccines, especially when compared with the whole-proteome level. The JYNNEOS vaccine induced weaker responses that correlated with the increased morbidity of the black-tailed prairie dogs when they were challenged with MPXV. The results regarding the efficacy of the JYNNEOS vaccine in this model agree with the results obtained by using the third-generation vaccine in the post-exposure model [138].

## 4. Stimulation of the Protective Immune Response—Long Memory against OPXV Infections

The ultimate success of vaccines depends on their evoking and maintaining immunological memory. How then, does a smallpox vaccine work? To answer this question, let us compare the infection process of the vaccines for OPXV (VACV) and VARV. First, the great success of the smallpox vaccine was based on the fact that it induced cross-protection. Second, VACV strains used in immune response stimulation, in contrast to the VARV strains that disseminate in the host with a very high viral load, remained localized in the site of inoculation, causing local infection limited to this area. Once it was introduced intradermally, it led to the development of a characteristic pustule (a ‘take’) at the vaccination site, which was considered evidence of protection [19,23]. VACV retention at the site of inoculation points to the role of the innate immunity, which controls systemic viral dissemination in the host, giving time for the development of antigen-specific adaptive immunity. VARV, during the first stages of infection, probably goes undetected by the mechanisms of innate immunity due to the presence of the genes that encode molecules responsible for the circumvention of the protective mechanisms and for rapid replication before adaptive immunity starts to control infection.

OPXVs are a group of viruses that replicate completely in the cytoplasm. Intracellular detection of viral infection is crucial not only to the induction of the innate immune response but also for the polarization of protective mechanisms of antiviral adaptive immunity. The recognition of cytoplasmic DNA is based on the presence of cytoplasmic DNA sensors and adaptor proteins. This includes pattern recognition receptors (absent in the melanoma [AIM-2] receptor and Toll-like receptor [TLR] 9), cyclic GMP-AMP synthase (cGas), the DNA-dependent activator of IFN-regulatory factors (DAI), DNA-dependent protein kinase (DNA-PK), IFN-γ inducible protein 16 (IFI16), the stimulator of interferon genes (STING), and RNA polymerase III, which converts poly(dA:dT) to poly(A:U)-rich dsRNA, which, in turn, serves as a RIG-like receptor agonist. Engagement of those sensors through binding cytoplasmic viral DNA induces the synthesis of host defense molecules such as type I interferons (IFN-α/β), II (IFN-γ), and III (IFN-λ1, -λ2, -λ3), proinflammatory cytokines, and chemokines (Figure 4). OPXVs possess a large genome that encodes a great number of proteins that can modulate the first steps of immune response induction. These mechanisms have been summarized in recently published articles [139,140,141].

Although viruses can escape the DNA-sensing system and/or inactivate sensors together with the downstream signal transduction pathways, secreted host proinflammatory mediators are still responsible for shaping the first steps of immune response, and that includes creating a proper cytokine and chemokine environment, among which interferons are the key antiviral molecules. Interferons show immunoregulatory properties affecting all cells presenting appropriate receptors for those mediators to elicit an immune response. They induce the synthesis of antiviral proteins that target viral entry, RNA expression, viral protein synthesis, viral assembly, and the release of viral particles from the infected cell. Inducing those mechanisms, IFN-γ protects neighboring cells against viral infection. Furthermore, IFN-γ signaling is involved in cell-mediated immune response induction, regulating and bridging innate and adaptive immunity [142,143].

Alternative animal and/or cell culture research models must be used to understand the relationship between the host and OPXVs during viral infection. The parallel animal model for VARV is ECTV infection, responsible for mousepox. Studies using the murine model have shed some light on the mechanisms involved in the viral evasion of host protective mechanisms. Antigen-presenting cells (dendritic cells, macrophages, B cells) play a crucial role in the induction of the adaptive immune response, firstly by recognizing the infection through the proper sensors, and secondly, by presenting the epitopes in the context of the major histocompatibility complex (MHC) class I or class II. Studies conducted *in vitro* on murine conventional dendritic cells (cDCs) challenged with ECTV have indicated that the virus abolished the ability of cDCs to stimulate CD4^+^ T cell proliferation required for the development of cell-mediated or humoral immunity, depending on the subpopulation of the CD4^+^ T cells. Furthermore, murine cDCs derived from bone marrow challenged with ECTV failed to induce, and even actively suppressed, the production of Th1 (IFN-γ, IL-2), Th2 (IL-4, IL-10), and Th17 (IL-17A) cytokines by CD4^+^ T cells. Furthermore, an *in vitro* challenge of JAW II DCs and RAW 264.7 macrophage cell lines with ECTV revealed that the virus affected the downstream pathway dependent on non-canonical NF-κB signaling, which is accountable, among others, for the stimulation of antiviral immunity. These data may suggest the impairment of the development of adaptive immunity in the first stages of ECTV infection [144,145,146]. These results are in accordance with data obtained by Rhem et al. from the VACV model, indicating the ability of the virus to modulate the first steps of adaptive immune response induction. A challenge of rat peritoneal macrophages, bone marrow-derived DCs, or RAW macrophage cell lines with the VACV Western Reserve strain was responsible for decreasing the level of MHC class II molecules as well as for the synthesis of chemokines, cytokines, and nitric oxide production. Moreover, VACV-induced apoptosis in those cell populations, in contrast to the B cell line, indicated that VACV can differently affect various antigen presenting cells. These data demonstrate that the impairment of antigen presenting cells’ critical functions is the OPXVs pathway for the evasion of innate immune protective mechanisms [144,147]. In addition, mice lacking the components of the complement system developed severe mousepox with increased mortality when challenged with ECTV [148]. Additionally, OPXVs possess genes that encode immunomodulatory proteins that impact the critical, intracellular pathways, and are responsible for affecting the host’s cytokine and chemokine environment, as well as for suppressing the production of antiviral proteins, including IFN-αbp blocking transmembrane signaling, B8R, a homolog of the cellular receptor for IFN-γ, B18R, a homolog of type I IFNs, E3L binding dsRNA and protecting against the activity of protein kinase R (PKR), K3L targeting PKR, and other proteins already presented [139,141,149,150,151,152,153,154,155]. However, in the case of VACV strains used as a vaccine, these strains cannot effectively evade the innate immune response, leading to the local replication of the virus. As a result, host protective immune response is induced [156].

Bioterrorism threats, the emergence of new zoonotic OPXV diseases, and waning anti-poxvirus immunity in the human population worldwide due to the cessation of SEP has brought back a very important question: how long does effective smallpox immunological memory last? To protect the host, both humoral and cell-mediated immune responses are required. Indeed, approved smallpox vaccines are powerful tools in the induction of both branches of the protective immune response.

Studies using animal models have provided an understanding of the adaptive immune mechanisms required to protect against secondary (recall) OPXV infections. The progress in the analysis of VACV-encoded proteins has allowed us to distinguish epitopes that are recognized and targeted for humoral and cell-mediated immune response after vaccination [157,158,159,160]. Moreover, data from those studies indicated that the majority of VACV strain epitopes recognized by host immune systems are found as conserved epitopes in VARV, and therefore are relevant to the immune defense against smallpox.

The cell-mediated immune response dependent on CD8^+^ T cells and CD4^+^ T cells as helper cells is induced after smallpox vaccination. It was confirmed that this type of immunity is required for viral clearance during primary OPXV infection [116,161]. Virus-specific CD8^+^ T cells control the spread of a virus in the host by direc killing infected cells and secreting antiviral factors and mediators involved in the recruitment of other immune cells to the site of infection. The hallmark of CD8^+^ T cell-secreted mediators are IFN-γ and IL-2, and they are the most commonly evaluated features of cell-mediated immunity.

The significance of CD8^+^ T cell-dependent immunity mediated by IFN-γ was presented in the model of respiratory infection of the VACV Western Reserve strain, where IFN-γ signaling was responsible for lung pathology reduction, the inhibition of virus dissemination, and the early clearance of the virus [162,163]. Also, the model with recombinant VACV expressing IFN-γ used for the post-exposure protection against both VACV and ECTV emphasized the role of this cytokine. Moreover, the studies of Xu et al. clearly showed that not only antibodies but also recall responses by long-lived memory CD8^+^ T cells could prevent clinical mousepox [164,165]. Preexisting antigen-specific CD8^+^ T cells, elicited using a VACV immunodominant epitope derived from viral IFN-γbp, encoded by VACV B8R gene—B8R_20–27_ (TSYKFESV), combined with CpG ODNs as adjuvants, cross-protected mice against ECTV challenge and limited the viral load during primary OPXV infection [132]. The cell-mediated immune response can block mortality in animals lacking B cells when they are challenged with VACV. Furthermore, CD8^+^ T cells can provide partial protection in the absence of a humoral immune response [116,161]. However, a CD4^+^ T cell population is required for the development of both cell-mediated and humoral immunity, since CD4^+^ T cell deficiency results in delayed viral clearance and increased mortality, as well as a lack of development of OPXV-specific antibodies [166,167].

Immunological VACV-specific memory responsible for maintaining critical protection against OPXV infections in humans is based on the functions of T cell populations. Studies done by Abate et al. revealed that effector memory CD4^+^ and CD8^+^ T cells, collected from VACV-specific vaccinees, were capable of antigen-specific proliferation upon *in vitro* stimulation, together with IFN-γ production. Additionally, the expansion of VACV-specific CD8^+^IFN-γ^+^ T cells correlated with their cytotoxic ability. Moreover, the memory effector γδT cells, which expressed proliferative ability and the potential to synthesize IFN-γ, were found among analyzed cells [168]. Similar results regarding the stimulation of γδT cells in subjects vaccinated with the recombinant vaccine based on the canarypox virus (CNPV), which expresses human immunodeficiency virus (HIV) proteins, were received. Lymphoproliferative and IFN-γ dependent immune response was induced by both CNPV and the HIV recombinant glycoprotein. However, only CNPV induced and increased the γδT cell-dependent immune response [169].

The direct and indirect cooperation between the innate and adaptive immune responses during viral infection leads to the control of viral dissemination in the host, viral clearance, and, finally, recovery. Antigen presenting cells play a crucial role in the induction of adaptive immunity dependent on the CD4^+^ and CD8^+^ T cells. However, the early steps of immune response during viral infection are fundamental for the subsequent course of infection. They include the production of proinflammatory cytokines responsible for the recruitment of neutrophils, monocytes, natural killer cells, and γδT cells, forming the first line of defense.

γδT cells create a unique population of lymphocytes that mature in the thymus ahead of αβT cells, subsequently migrating, in controlled waves, from the thymus to organs and tissues such as the lungs, gut, skin, uterus, and tongue, forming several subpopulations of γδT cells with distinct functions. It should be emphasized that γδT cells do not require antigen presentation in the context of MHC molecules for their stimulation, in contrast to CD4^+^ and CD8^+^ T cell activation. Induction of γδT cells leads to cytokine production and the proliferation and stimulation of their cytotoxic activity. In addition, some of the subpopulations of γδT cells can work as APCs themselves, inducing adaptive immunity [170,171,172,173,174].

The relevance of γδT cells during OPXV infections was demonstrated by the assessment of several γδT cell subpopulations in different animal models. Common results, irrespective of the animal model and the evaluated γδT cell subpopulation, indicated the very early involvement of those cells during the viral challenge and profound IFN-γ production. One of the murine γδT cell subpopulations is a subpopulation of dendritic epidermal T cells (DETCs) localized in the skin, the portal of entry for many infectious agents, including OPXVs. The involvement of DETCs during intradermal ECTV challenge was assessed, indicating a rapid increase in the number of DETCs within 24 h post-infection. After that, the number of DETCs decreased, reaching control values within the next 24 h. Moreover, fast but short-lasting IFN-γ secretion by purified DETCs was observed, together with a correlation between the expression of genes encoding *IFN-γ* and *CCL5*, the chemokine responsible for macrophage and neutrophil attraction. This was also accompanied by the expression of genes encoding immunoregulatory factors, such as TGF-β, GM-CSF, and KGF, important for maintaining skin integrity. DETCs from mice infected with ECTV were rapidly induced to secrete mediators that contributed to both immune protection and the control of skin integrity [175]. A similar engagement of γδT cells was presented in a VACV murine model, although the assessed γδT cells were derived from the peritoneal cavity and spleen. C57BL/6 wild-type (WT) mice and β TCR knockout (KO) mice were challenged with VACV, resulting in a rapid and intense increase in IFN-γ producing γδT cells in both strains of mice as early as on the second day post-challenge. Moreover, they expressed cytolytic activity and were responsible for the early clearance of the virus, since in γδ TCR KO mice the VACV titer in the spleen, liver, and fat pads was 2.5–3.0 fold higher than in WT animals [176].

Recent studies by Dai et al. proposed the mechanism for the involvement of splenic γδT cells in the promotion and regulation of antigen-specific CD8^+^ T cell response during VACV challenge [148]. Their data are in accordance with data presented by Selin et al. regarding the early engagement of γδT cells in proliferation and IFN-γ production, prior to the activation of VACV-specific CD8^+^ T cells [147]. Moreover, the adoptive transfer of splenic γδT cells to δ TCR KO mice challenged with VACV controlled the viral load, as well as it induced a the higher level of VACV-specific and IFN-γ producing CD8^+^ T cells compared to δ TCR KO mice only challenged with the virus. The proposed basis of this phenomenon was the engagement of γδT cells working as APCs, presenting VACV epitopes in the context of MHC class I, to CD8^+^ T cells. Triggering the intracellular pathway dependent on MyD-88 signaling was required for the induction of γδT cells with VACV since γδT cells derived from WT mice, adoptively transferred into MyD-88 KO mice challenged with VACV, were sufficient to rescue γδT cell expansion and IFN-γ secretion [177].

γδT cells, due to their prompt involvement upon viral infection, are the major source of IFN-γ, a key molecule involved in protective immunity against intracellular pathogens, and thus are responsible for shaping the subsequent events of innate and adaptive immune response. Due to the ability of γδT cells to function as APCs, they present viral epitopes in the context of MHC class I, and by the increased level of costimulatory molecules, they deliver the required signals (1, 2, and 3), influencing naïve CD8^+^ T cell activation. γδT cells function as the bridge between innate and adaptive immunity, regulating the antigen-specific T cell immune response [142,168,169,175,176,177].

The relevance of neutralizing antibodies was established since the presence of humoral immunity correlated with the protection against smallpox and is thought to be the best laboratory predictor of protective immunity to OPXV infections in humans [178]. The individuals with a low titer of neutralizing antibodies were prone to develop a ‘take’ after re-vaccination, indicating the requirement of antibodies to combat secondary infection [179]. Historical reports of cases, in which the passive transfer of serum from VARV- or VACV-immune individuals protected exposed subjects against smallpox, are available [180]. Studies using animal models confirmed the significance of protective humoral immunity. Chaudhri et al., utilizing the ECTV-C57BL/6 mice model, indicated that persistent ECTV infection resulted from the deficiency of B cells or CD4^+^ T cells (due to the deficiency in MHC class II), despite the presence of functional antigen-specific CD8^+^ T cells [181,182]. Murine models of OPXV infections indicated important roles for both cell-mediated and humoral immunity. However, studies conducted on rhesus monkeys using the MPXV model revealed that the depletion of CD4^+^ T cells or CD8^+^ T cells from VACV-immune animals prior to the infection with MPXV did not reduce protection against lethal challenge, which was in opposition to the depletion of B cells. Animals that were depleted of CD20^+^ B cells at the time of vaccination to block the neutralizing antibody formation, died when the MPXV challenge followed. To confirm the involvement of neutralizing antibodies, the researchers administered human vaccinia immune globulin (VIG) to unvaccinated monkeys 4 days before the challenge with MPXV. All animals developed skin lesions, but most importantly, none of the monkeys died. This shows the importance of neutralizing antibodies in the protection against lethal infection [167].

Studies conducted on murine models confirmed the importance of both branches of the adaptive immune response to combat OPXV infections. Using knock-out models for VACV or ECTV challenge, it was demonstrated that mice deficient in B cells as well as in β2-microglobulin were protectively vaccinated. Although mice with a decreased number of CD4^+^ T cells or MHC class II knock-out and double knock-out, both MHC class I- and MHC class II-restricted activities, were either poorly protected or unprotected. This indicates that neutralizing antibodies and antigen-specific CD8^+^ T cells yield overlapping modes of vaccine protection in mouse models [13].

### Immunological Memory against OPXV Infections

Since the last natural case of smallpox was recorded in Somalia in 1977, and since the global vaccination program ended in 1980, the question posed by scientists concerned the longevity of immune memory against smallpox induced either naturally or through vaccination. Cohort studies were conducted and focused on the evaluation of the long-lived plasma cells and memory B cells in the blood of smallpox vaccine recipients (Figure 5). The blood samples collected from individuals vaccinated between 3 months and 60 years earlier revealed the presence of not only antigen-specific B cells but also neutralizing antibodies. The kinetics of virus-specific memory B cells indicated an initial decline post-vaccination, then reached the plateau ten times lower than the peak, and then established a stable level for more than 50 years. Additionally, there were no significant differences in memory B cell levels in the individuals vaccinated between 20 and 60 years earlier. Moreover, neutralizing VACV-specific antibodies were detected in all blood samples, with no indication of decline between 1 and 60 years. The recall response of memory B cells induced by the revaccination of the tested subjects, who had been vaccinated 22 and 48 years earlier, allowed the solving of the issue of the functionality of virus-specific memory B cells. The anamnestic response in the revaccinated individuals induced a 20-fold increase in antibodies. The antibodies were directed towards VACV integral membrane protein H3L, which was identified as a target for the human humoral immune response and is probably involved in the adsorption of the virus to the target cell surfaces [183,184]. Studies done by Hammarlund et al. provided new insight into protective humoral immunity. Humoral immunity in evaluated individuals showed the same range of neutralizing antibodies observed between 1 and 7 years after smallpox vaccination. Moreover, neutralizing antibodies were observed for up to 75 years post-vaccination. These data indicate lifetime humoral protective immunity after a single smallpox immunization [185]. Additionally, high titers of neutralizing antibodies were detected in unvaccinated survivors of VARV infection who had caught the disease more than 40 years earlier [186].

An evaluation of the specificity and activity of neutralizing antibodies resulting from immunization by smallpox vaccines suggested that a protective mechanism is induced during OPXV infection. Immunization with the smallpox vaccine elicited antibodies specific for antigens of both MV (H3, A27, D8, and L1) and EV (B5 and A33) infectious forms of OPXVs. Proteins that are described as attachment and entry proteins play crucial roles during OPXV infection. Moreover, the presence of complement increased the inhibitory activity of those antibodies. Since the sequence homology among the surface proteins from VACV, CPXV, MPXV, and VARV is very high, it explains the efficient cross-protection elicited by the VACV vaccine [4,187,188].

Similar studies were launched to assess the presence and functionality of virus-specific CD4^+^ T cells as well as CD8^+^ T cells in recipients of smallpox vaccines or survivors of VARV infections. Cohort studies evaluated the functionality of those T cell populations using ELISA or intracellular staining for IFN-γ and TNF-α detection, as well as their proliferative abilities upon *in vitro* restimulation with VACV. CD4^+^ T cells, with the characteristics of the memory phenotype IFN-γ^+^TNF-α^+^, were observed 1 year and 61 years after vaccination. Although the frequency of virus-specific CD4^+^ T cells slowly waned over time, this population could be detected as late as 75 years post-immunization. Similarly, CD8^+^ T cells collected from vaccinees’ blood responded to VACV *in vitro* restimulation with the ability to produce both cytokines, IFN-γ and TNF-α. It seemed that the memory based on virus-specific CD8^+^ T cells waned faster compared to that based on CD4^+^ T cells, with an estimated half-life of 8–15 years, but those cells were detected starting at 27 days post-vaccination up to 61 years post-immunization [185]. These data were confirmed by Combadiere et al. in their cohort studies of individuals vaccinated with the smallpox vaccines once or repeatedly. The frequency of IFN-γ producing T cells with the ability to proliferate upon *in vitro* restimulation with VACV was decreased depending on the time of the first vaccination. The methods used for evaluation allowed the distinction, in expanded *in vitro* assessed cells, of the presence of VACV-specific proliferative responses in 72.5% of vaccinees. A simultaneous analysis of the samples revealed, firstly, the presence of rapid IFN-γ−producing effector-memory T cells, secondly, VACV-specific memory cells without the ability to produce IFN-γ, and thirdly, the population with neither effector-memory nor proliferating memory T cells. The number of recalls did not influence the functionality of the evaluated cells. However, revaccination of the individuals induced intense effector-memory and proliferative VACV-specific immune responses. Moreover, revaccination of those subjects who had no detectable IFN-γ-producing cells during the first time of evaluation stimulated a measurable potent immune response within 2 months of vaccine recall. Survivors from VARV infections had low levels of circulating CD4^+^ T cells responding to VACV antigen stimulation up to more than 40 years after infection. These data show the mechanisms of the persistence of long-memory against OPXVs in the absence of antigens [186,189].

It seems that, in humans, both humoral and cell-mediated immune responses are required to combat OPXV infections. Individuals with immunodeficiencies in terms of either B cells or T cells are at higher risk of complications after vaccination with first- and second-generation smallpox vaccines. Now, we understand that T cell deficiency has a greater impact on the patient’s health than agammaglobulinemia has. The loss of B cell immunity leaves the T cell-dependent branch of adaptive immunity relatively intact. Severe impairment of T cells, especially CD4^+^ T cells, has an immediate impact on the antiviral immune response, including antibody production and the development of cell-mediated immunity. Such manifestations after smallpox vaccination were demonstrated in cases of vaccinia necrosum, a life-threatening complication. Although vaccinia necrosum is associated with defects in cell-mediated immunity, 100% (18 of 18 cases) of patients were unable to mount sufficient levels of neutralizing antibodies. This condition would have been fatal in 100% of cases if VIG therapy had not been introduced. The adoptive transfer of immune-competent T cells from smallpox-vaccinated donors cured one case of vaccinia necrosum that did not respond to VIG therapy. This implies models of either humoral or cell-mediated immunity can be important against lethal OPXV infections [180,185].

The importance of cell-mediated protective immunity was presented in both animal models and survivors of VARV infection. Revaccination of VACV recipients up to 4 days post-VARV exposure was accountable for the reduction of mortality and morbidity from smallpox [23]. Currently, the majority of the human population presents no immunity toward OPXV infections. The strategies used to combat the potential outbreaks rely on the rapid induction of OPXV-specific immunity induced in a post-exposure period. The controlled assessment of cell-mediated and humoral immunity stimulated by the smallpox vaccine was evaluated in previously vaccinated (VACV-non-naïve) and VACV-naïve individuals. Both groups were vaccinated with Dryvax, then the IFN-γ-producing T cells, lymphoproliferation, and the production of VACV-specific antibodies were evaluated. The obtained data indicated that both groups developed cell-mediated and humoral immunity. However, VACV-non-naïve individuals’ cell-mediated immunity (detected 7 days after vaccination) corresponded with, and in some cases, preceded, antibody response by more than a week. In VACV-naïve subjects, T cell-dependent immunity developed ahead of the neutralizing antibodies. These data suggest that protective immunity after re-exposure to OPXVs may rely on T memory cells [190].

The development of prophylactic measures against OPXV infections, especially against the current threat caused by MPXV, requires an understanding of the immunobiology of OPXVs as well as knowledge regarding the interplay between viruses and immune effector host cells. Additionally, the evaluation of the immune response induced by the VACV vaccine provided information regarding the longevity and functionality of stimulated antiviral mechanisms.

## 5. Conclusions

Herd immunity against smallpox has been steadily declining since the cessation of vaccinations in 1980 through the global eradication program launched by WHO in 1966. Smallpox vaccines can confer a level of cross-protection against other OPXV infections. Repeated OPXV outbreaks have been reported worldwide as the result of animal-to-human transmission, especially in young persons as well as those with immune deficiencies. Now, the median age of those patients has increased, which correlates with the waning immunity against VARV caused by the cessation of mass smallpox vaccination. The awareness of the possibility of OPXV infections should be greater among medical professionals and veterinarians since the early diagnosis and isolation of infected animals can curtail the spread of the virus. Furthermore, clinical signs of OPXV infections were often confused with the outcomes of bacterial infections.

Monkeypox is a zoonotic disease predominantly reported in Central and West Africa. Recently, a rapid increase in monkeypox cases was reported outside of the endemic area of MPXV. Moreover, the number of diagnosed MPXV infections, probably underestimated due to the limited diagnostic capacity in many regions, indicates that the virus spreads in the human population through direct human-to-human contact, which poses a risk to global public health. There are many questions about the present outbreak, as well as many factors that influenced the spread of MPXV outside of its endemic area, such as human encroachment into wild habitats, the uncontrolled transportation of animals, human migration, and the increased frequency of travel. Did the waning of herd immunity against smallpox, together with those factors, facilitate the MPXV outbreaks? Probably, yes. However, the confirmed number of monkeypox cases suggested the new mechanisms of MPXV transmission. It is still not known if mutations in MPXV could have been responsible for making the virus more human-transmissible. Further research focusing on the analysis of MPXV genomes from the current outbreaks should be conducted to clarify this issue. Nevertheless, monkeypox is a disease that should be treated as any other zoonosis. This means that, to control the disease, early recognition of infections, good hygiene practices, and self-isolation or quarantine for the patients and their contacts should be applied, as well as vaccination, which is probably the best way to control OPXV infections. In the current situation, WHO does not recommend global vaccination against MPXV; however, post-exposure vaccination will induce protection against the disease. The lesson from COVID-19 is that the emergence of an animal infectious agent can become a global health threat, and we need to be prepared.

The cohort studies performed on survivors of smallpox and those vaccinated against the disease provided information regarding the longevity and functionality of stimulated antiviral mechanisms, which lead to effective protection. Moreover, vaccination with the smallpox vaccines protected against MPXV infection by up to 85%. Developing prophylactic measures against OPXV infections, especially against the current threat caused by MPXV, requires an understanding of the immunobiology of OPXVs as well as knowledge regarding the interplay between those viruses and host immune effector cells. Since working with VARV is highly restricted, animal and cell line models, as well as other OPXVs, have provided insight into the defense mechanisms of host antiviral protection, as well as OPXV evasion pathways. To survive within a host, OPXVs use many immunomodulatory proteins. A circumvention of viral evasion strategies and an enhancement of the major host defenses are key in designing novel, safer vaccines, and should become the goal for antiviral therapies treating OPXV infections.

## Figures and Tables

**Figure 1 pathogens-12-00363-f001:**
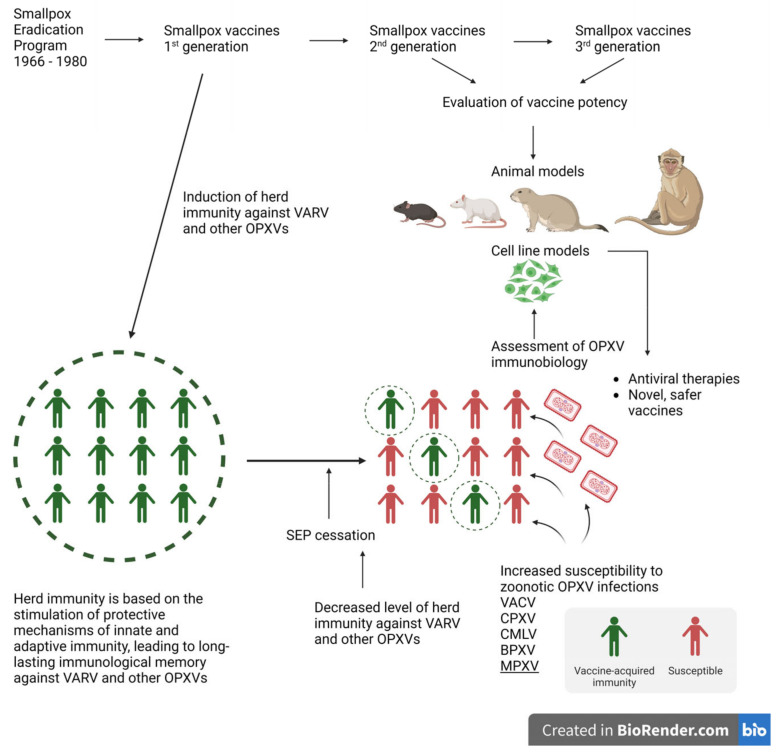
The graphic aim of the review. The figure was created with the BioRender application.

**Figure 2 pathogens-12-00363-f002:**
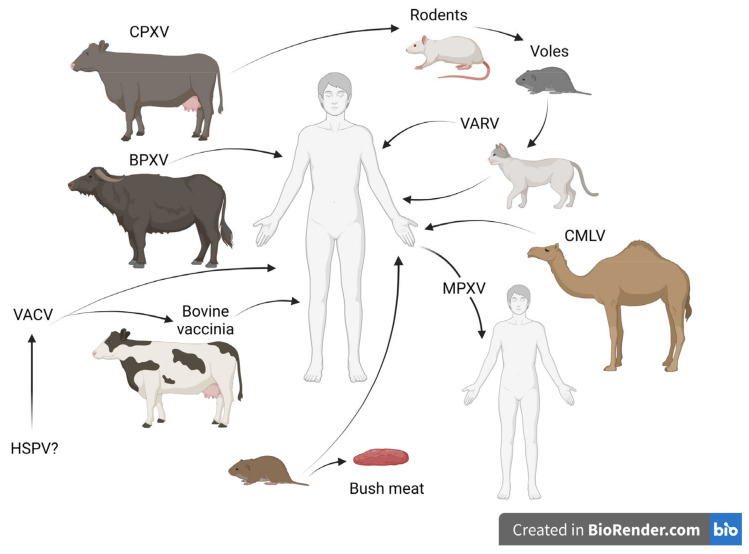
Schematic representation of OPXV transmission to humans indicating potential reservoirs of the virus. Transmission of zoonotic OPXVs occurs through direct contact with infected animals as well as with their body fluid. This mechanism also occurs in MPXV infection, although the reservoir has yet to be clarified. Viral transmission is the result of direct contact with the animals, and by handling, preparing, and consuming wild game. Human-to-human MPXV transmission is the result of direct contact with patients, their lesions, or body fluids, as well as the crust materials. VARV can be transmitted human-to-human *via* the respiratory route (sputum and saliva), as well as by direct contact with the skin, body fluids, and ulcers/scabs. The figure was created with the BioRender application.

**Figure 3 pathogens-12-00363-f003:**
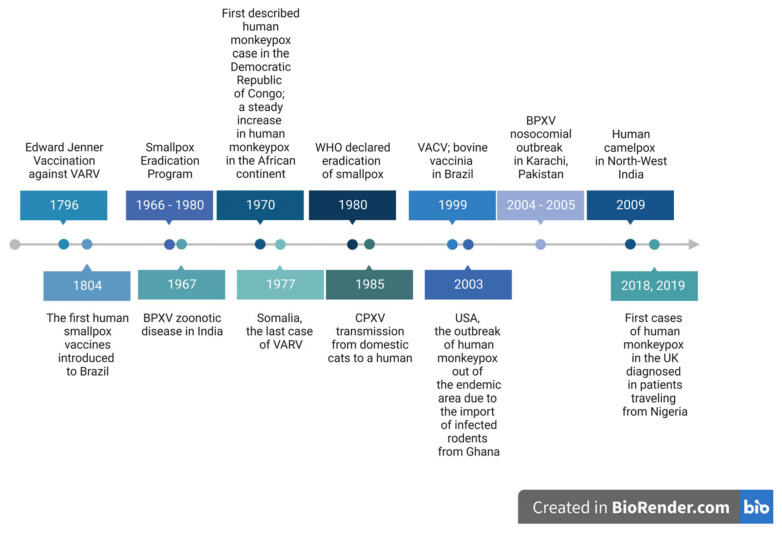
A horizontal timeline of main events and OPXV outbreaks. The figure was created according to the references included in the review using the BioRender application.

**Figure 4 pathogens-12-00363-f004:**
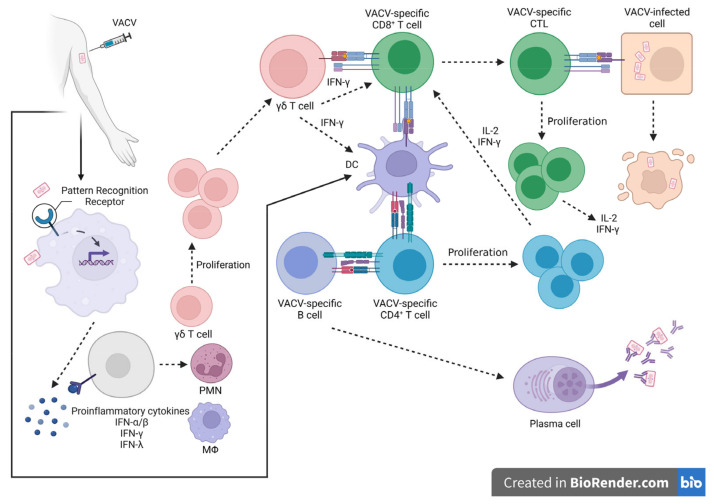
Stimulation of protective immunity against OPXVs caused by vaccination with VACV. The induction of the protective immune response begins with the recognition of viral particles through pattern recognition receptors, leading to the synthesis of type I, II, and III interferons, as well as the synthesis of proinflammatory cytokines and chemokines. These cause the recruitment of neutrophils, natural killer cells, monocytes, γδT cells, and the activation of antigen-presenting cells (APCs), which are crucial for the induction of adaptive immunity. APCs, by capturing and processing viral antigens, present their epitopes in the context of the major histocompatibility complex (MHC) class I or class II, stimulating CD8^+^ T cells and CD4^+^ T cells, respectively. CD4^+^ T cells, upon stimulation, secrete cytokines that polarize the immune response towards the humoral (B cells) and cell-mediated (CD8^+^ T cells) immune responses. The help delivered by the CD4^+^ T cells to the B cells through direct contact and cytokines (mainly IL-4, IL-5, IL-13) results in the transformation of the B cells into plasma cells that secrete VACV-specific antibodies, and long-living memory B cells. Cell-mediated protective immunity depends on the stimulation of CD8^+^ T cells through direct contact with APCs. The major population of APCs responsible for stimulation of the primary cell-mediated immune response are dendritic cells and the recently elucidated γδT cells that can serve as APCs. CD8^+^ T cells, upon direct cell-to-cell stimulation and through cytokine (mainly IL-2, IFN-γ) indirect stimulation, form VACV-specific cytotoxic T cells and long-lived memory CD8^+^ T cells. The figure was created with the BioRender application.

**Figure 5 pathogens-12-00363-f005:**
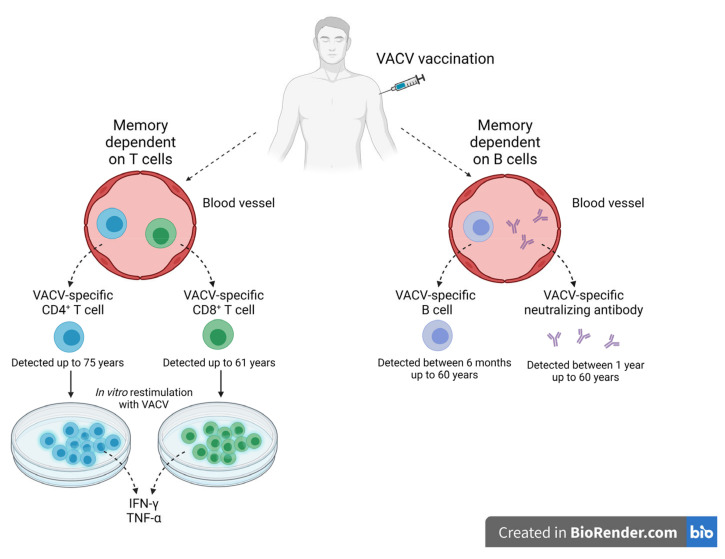
The longevity of immunological memory against VACV—the result of vaccination. Immunological memory against VACV was evaluated through cohort studies assessing the presence, specificity, and functionality of memory B cells, as well as memory CD4^+^ and CD8^+^ T cells. VACV-specific B cells and VACV-specific antibodies were detected in the blood of vaccinated individuals up to 60 years post-vaccination. Memory CD8^+^ T cells and memory CD4^+^ T cells were detected in the blood of vaccinated individuals up to 61 and 75 years post-vaccination, respectively. Both cell populations, upon *in vitro* stimulation with VACV antigens, proliferated and secreted IFN-γ and TNF-α, indicating their functionality. The figure was created with the BioRender application.

## Data Availability

Data Availability Statements are available in section “MDPI Research Data Policies” at https://www.mdpi.com/ethics (accessed on 28 December 2022).

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
