# Peer review of "Orthopoxvirus Zoonoses—Do We Still Remember and Are Ready to Fight?"

_pathogens, 2023, doi:10.3390/pathogens12030363_

Round 1

Reviewer 1 Report

Małgorzata and colleagues presented a review of the literature on Orthopoxvirus zoonoses and transmission. In this review, authors have presented features regarding the orthopoxvirus's zoonotic infections, factors responsible for viral transmissions, and the emerging problem of the increased number of recently reported monkeypox cases. Also, discussed the development of prophylactic measures against poxvirus infections, with a major focus on the current threat caused by the monkeypox virus, requires a profound understanding of the poxvirus immunobiology. Further, the authors discussed the utilization of animal and cell line models used in host antiviral defenses, as well as the orthopoxvirus evasion mechanisms, including a large number of proteins that subvert inflammatory, evasion strategies, and immune pathways.

This critical and timely review presented insights into designing novel, safer vaccines that facilitate antiviral therapies in treating poxvirus infections and immunopathology.

The major drawback is that the authors have extensively presented the text but not summarised get literature in the cartons, so I suggest the authors revise the review with the cartoons/tables showing the following text.

1-    Graphic covering the entire AIM of the review

2-    Phylogenetic tree

3-    Table showing outbreaks caused by poxviruses

4-    Prophylactics and mechanisms -a cartoon that includes stimulation of the protective immune response – long memory against OPXV infections.

5-    Cartoon showing immunological memory

Author Response

Reviewer 1

Thank you very much for your comments. We really appreciate them and agree that for the improvement of the manuscript quality the figures were required.

- The graphic aim of the review was prepared.

- The phylogenetic tree was not included, however, additional information regarding the availability of the current OPXVs phylogenetic tree was added.

- Instead of the table showing outbreaks caused by poxviruses, a figure displaying a timeline with chronological important achievements and major outbreaks was included.

- The figures presenting stimulation of the immune response and formation of immunological memory were added

Reviewer 2 Report

In the title of their review article on orthopoxvirus zoonosis the authors raise the intriguing question if we still remember and are ready to fight? The review begins with an introduction from the angle that a better understanding of immunity is beneficial for vaccine development with a historic spin on variola virus and the eradication of smallpox. The authors elaborate on zoonotic infections describing several orthopoxviruses and related diseases at varying detail before focusing on monkeypox prophylaxis due to the ongoing outbreak. A lengthy section on protective immune responses and memory against orthopoxvirus infections follows before the authors conclude by stating the obvious, i.e. that awareness for poxvirus infections should be greater among medical professionals, mutations in monkeypox virus could have changed transmission routes and that circumvention of viral evasion strategies is the key to designing novel vaccines.

Unfortunately, the authors stay at a largely superficial level in their review and do not go into the cellular or molecular details of virus-host interactions. The review reads as an accumulation of sentences that convey a lot of information paraphrased from the literature without actually citing the sources where appropriate. Rather, several citations are accumulated at the ends of paragraphs. The authors frequently use vague descriptions like “enormous size”, “high similarity” or “many proteins” instead of providing quantitative information even though it is available in the primary literature. Furthermore, it appears as if the authors for large parts copied lists of viruses or proteins from other review articles without citing original research. While the authors write lengthy sections on the disease burden caused by orthopoxviruses the reader is never presented with any quantitative information on factors responsible for viral transmission. The concluding statement “monkeypox is a disease that should be treated as any other zoonosis” suggests willful ignorance at best. Sadly, it is never revealed if we still remember or are ready to fight but instead the design of novel, safer vaccines for treating orthopoxvirus infections is recommended. Fortunately, many research groups focus their work on exactly this, even though the authors prefer not to name names or disclose where they gained this information.

The review article should not be published in its current form. It could, however, be improved by placing references where they belong and including primary literature rather than reviews throughout the text. The following points should be addressed:

75: Which 6 proteins? Vague.

100: What do the authors mean with “certain impossible achievements in last XX century”?

122: Wrong definition. Not all zoonoses represent a threat to global health.

125: What several others?

138: As described in the cited references this is not the definition of a virulence gene but what they are referred to.

149: A source to document the increasing prevalence of poxvirus incidents is missing. Which other factors are the authors referring to?

158: Which zoonotic poxviruses were introduced in this manner? Where did this happen? What is the source for this statement?

177: Source missing!

247: Factually wrong. See e.g. https://doi.org/10.1016/j.vetmic.2009.08.026 Also, the authors contradict their own statement in line 155 regarding spread of poxviruses occurring in close contact with animals and humans.

263: Unspecific. How many cases are the authors talking about? How much did these increase?

269: The statements are misleading. All OPXV carry host immune evasion genes and phylogenetic closeness itself does not raise public health concerns.

287: Was is not the other way around? The virus was first sequenced and then found to be VACV derived?

311-317: This list was copied from reference [42] without referring to the primary literature. Also, it should be stated that these features are not unique to CMLV.

333: What is the natural host?

450: Why change style completely and name first authors here?  

468: What is the supposed difference between novel, prophylactic or therapeutic vaccines? This does not make sense.

471: In which respect were vaccines improved?

486: Which groups focus their work on the development of smallpox vaccines?

556: VARV is not the wt OPXV!

566: Speculative and without a source. Which genes?

582: Since when does genome size correlate with immune modulation?

584: Wrong referencing.

626: Like which proteins?

630: Logically and factually wrong.

778: References missing.

927: Could you name an example of an OPXV surviving in the host? Which proteins are used?  

Author Response

Reviewer 2

Thank you very much for your comments. We do believe that based on them our manuscript will meet the standards set by Pathogens.

- Line 75: The proteins were included with the appropriate references.

- Line 100: This statement stresses the importance of advanced molecular biology techniques, their input into novel vaccine design, and the possibility of using them in the laboratory construction of an infectious agent.

- Line 122: Thank you for your comment, that is true that not all zoonoses present a worldwide threat however, they all may become a significant health problem. The sentence was corrected: Zoonosis is an infectious disease that is naturally transmitted between species, from vertebrate animals to humans, and may present a significant risk to public health.

- Line 125: Thank you for your comment, the sentence was corrected.

- Line 138: Thank you for the comment, the sentence was corrected.

- Line 149: That is true, however, the problem of spreading monkeypox among homosexual men and stigmatization of the disease is not covered by this review, although the concern of healthcare professionals in Brazil and other American countries is understandable

- Line 159: The following references were included: https://www.cdc.gov/poxvirus/monkeypox/veterinarian/african-rodent-ban.html, Infectious diseases. U.S. monkeypox outbreak traced to Wisconsin pet dealer, An Increasing Danger of Zoonotic Orthopoxvirus Infections.

- Line 177: The following reference was included: https://www.cdc.gov/poxvirus/monkeypox/veterinarian/monkeypox-in-animals.html

- Line 247: Thank you for your comment, the appropriate fragment was corrected.

- Line 263: Thank you for your comment, however, since cases of CPXV in cats are often by many authors considered unrecognized, the exact number of those events is not given in our manuscript.

- Line 269: Thank you for the comment. It requires further explanation. Since CMLV and VARV are phylogenetically very closely related, and also VARV is considered non-existing in nature if CMLV carries genes related to immune evasion it may facilitate the reintroduction of VARV – hypothetically, but it is of public health highest concern.

- Line 287: That is correct, sequencing of BPXV has undoubtedly confirmed that it is derived from VACV Lister strain, however, in 1971, an article was published presenting such a conclusion based on traditional laboratory methods, doi: 10.1007/BF01249754 “Characteristics of a new poxvirus isolated from Indian buffaloes”; the reference was included in this sector”.

- Lines 311-317: Thank you for the comment, appropriate references were included.

- Line 333: Thank you for your comment, however, this issue was further explained.

- Line 450: The style of citing references was corrected, thank you.

- Line 471: Thank you for the comment; the 2nd generation smallpox vaccines improved in regards to standardization and control of preparation.

- Line 486: Thank you for the comment. It meant that many scientific groups focus their work on the immunobiology of OPXVs assessment and obtained data can be used in designing novel vaccines. The sentence was rephrased.

- Line 556: Thank you for your comment. However, VARV does not have a reservoir in nature like other OPXVs, therefore each infection with VARV should be considered as an infection with the wild strain/type of the virus. Even VARV deposits cannot be regarded as laboratory strains.

- Line 566: Thank you for the comment. Yes, we agree that it is speculative on some level. Although, the presence of the OPXV genes responsible for the evasion of the host protective mechanisms was proved by scientists and discussed earlier in the manuscript.

- Line 582: Thank you for your comment; that is true the size of a genome does not necessarily correlate with the ability of the virus to modulate immune responses; the sentence was corrected.

- Line 584: Thank you for your comment; it was corrected.

- Line 626: Thank you for your comment; this is general information regarding the ability of OPXVs to circumvent protective mechanisms. It will not improve the quality of this review if all viral proteins involved in the OPXVs immune evasion are enlisted. However, examples of these important viral proteins are included.

- Line 630: Thank you for your comment. This comes from work on VACV used as a vector for some human vaccines; it has been concluded that deletion of genes responsible for immune evasion results in replication-competent, greatly attenuated VACV, able to induce a protective immune response against OPXVs challenge.

- Line 778: This is the title of the paragraph.

- Line 927: Thank you for your comment. As long the virus replicates within the host it can be considered a survival of virus.

Reviewer 3 Report

This article is an ode to immunology and the history of Orthopoxvirus vaccines. The work is significant because the most efficient vaccine in history and the eradication program must be remembered. Virus studies in general are currently in short supply, and articles like this one can provide a brief summary of important information. Corrections: line 212-2013: please clarify the reference to this information. line 344: close the parentheses. Suggestion: line 226-229: It is important to mention the current conditions of the immunization program in Brazil, one of the best in the world, and the destruction provided by the last government (2018-2022) that was orchestrated in Brazil. Paragraph:495-498: This part of the text can be improved.

Author Response

Reviewer 3

Thank you very much for your comments. We really appreciate your input in the improvement of the manuscript. Please find out the corrections that were made.

Line 212-213: please clarify the reference to this information. Thank you for the remark, the reference has been clarified “Frequency of adverse events after vaccination with different vaccinia strains”; doi:10.1371/journal.pmed.0030272

Line 344: close the parentheses. It has been done.

Suggestion: lines 226-229: It is important to mention the current conditions of the immunization program in Brazil, one of the best in the world, and the destruction provided by the last government (2018-2022). that was orchestrated in Brazil.

Thank you for the suggestion. The proper information together and additional references were introduced (“Monkeypox in Brazil between stigma, politics, and structural shortcomings: have we not been here before?”; doi: org/10.1016/j.lana.2022.100394, “Twenty years after Bovine Vaccinia in Brazil: where we are and where are we going?”; doi: 10.3390/pathogens10040406, “The Brazilian National Immunization Program:46 years of achievements and challenges”.

Moreover, additional information regarding serological surveys and possible an alternative route of VACV spreading were introduced together with appropriate references: “Seroprevalence of orthopoxvirus in an Amazonian rural village, Acre, Brazil”; doi: 10.1007/s00705-010-0675-3, “Survival of Vaccinia virus in inoculated cheeses during 60-day ripening”; doi.org/10.3168/jds.2017-12560, “One more piece in the VACV ecological puzzle: could peridomestic rodents be the link between wildlife and bovine vaccinia outbreaks in Brazil?”; doi:10.1371/journal.pone.0007428

Paragraph:495-498: This part of the text can be improved. It has been corrected.

Round 2

Reviewer 1 Report

The authors have addressed the concerns and improved the review article significantly.

Author Response

Thank you for accepting the revised version of the manuscript

Reviewer 2 Report

I appreciate the author’s effort to improve the manuscript, especially including figures and rewriting entire passages as well as placing many references in juxtaposition to the corresponding statements. In some parts (e.g. 2.4 Monkeypox) the references are still aggregated at the end of paragraphs which makes reading the review difficult and I wish the authors had paid more attention to details  prior to submission. My specific concerns have been answered even though I disagree on the use of wt vs vaccine virus (see below). Unfortunately, my general comments were not heeded and the conclusion remains vague with statements I do not support. Nevertheless, I am ok with the review for a careful read and correction of several typos by the authors.  

150: the sentence is missing a verb

603: Thank you for your answer. However, I disagree on the use of terminology here. Wild type (wt) strains of a virus are historically those occurring in nature but the term is typically used for the most frequent genetic composition of a certain virus, not the genus. VACV, CPXV, MPXV, VARV, etc. all have a wt strain (and possibly many additional variant strains) within the genus orthopoxvirus (OPXV). But none of them is the wild type for the genus. Furthermore, they may all be used to illicit immunity against further poxvirus infection. Even VARV was used to protect people from smallpox (via variolation) before the discovery of vaccination.

873: “to succumb to a disease” typically infers the worst possible outcome which for smallpox is death. To avoid misunderstanding I recommend rephrasing “who caught the disease”

Author Response

Thank you for your comments

Line 150: The sentence was corrected

Line 873: The sentence was corrected

The typographical errors were corrected

Thank you for your comment. We agree with your taxonomic consideration. However, by comparing the process of host/human infection with vaccine OPXV (VACV), and wt OPXV (VARV), we wanted to show the difference between well adapted (VARV), and “foreign/new virus” for a host. Those differences are related to genetic events, suitable for avoiding host immune responses. In that meaning, VARV was considered as a “wt virus” that has elaborated throughout long-term evolution, strategies to avoid human immune responses in smallpox. By this VARV could have survived in human populations for thousand years, till Jenner’s time.

Elde et al. Cell 2012, 150(4), 831-841, doi.10.1016/j.cell.2012.05.049 presented elegant studies on the poxviruses’ recombination-mediated gene expansion enabling them to adapt fast to a new host species. Large genome with low mutation rates and the increased copy number of a gene(s) encoding factor(s) directly related to enhanced virus replication and inhibition of the host protein(s) involved in defenses. We corrected the sentence.